# Effects of Virgin Olive Oil on Blood Pressure and Renal Aminopeptidase Activities in Male Wistar Rats

**DOI:** 10.3390/ijms22105388

**Published:** 2021-05-20

**Authors:** Germán Domínguez-Vías, Ana Belén Segarra, Manuel Ramírez-Sánchez, Isabel Prieto

**Affiliations:** 1Unit of Physiology, Department of Health Sciences, University of Jaén, Las Lagunillas, 23071 Jaén, Spain; asegarra@ujaen.es (A.B.S.); msanchez@ujaen.es (M.R.-S.); 2Department of Physiology, Faculty of Health Sciences, Ceuta, University of Granada, 18071 Granada, Spain

**Keywords:** renal–aminopeptidase activities, systolic blood pressure, high-fat diet, virgin olive oil

## Abstract

High saturated fat diets have been associated with the development of obesity and hypertension, along with other pathologies related to the metabolic syndrome. In contrast, the Mediterranean diet, characterized by its high content of monounsaturated fatty acids, has been proposed as a dietary factor capable of positively regulating cardiovascular function. These effects have been linked to changes in the local renal renin angiotensin system (RAS) and the activity of the sympathetic nervous system. The main goal of this study was to analyze the role of two dietary fat sources on aminopeptidases activities involved in local kidney RAS. Male Wistar rats (six months old) were fed during 24 weeks with three different diets: the standard diet (S), the standard diet supplemented with virgin olive oil (20%) (VOO), or the standard diet enriched with butter (20%) plus cholesterol (0.1%) (Bch). Kidney samples were separated in medulla and cortex for aminopeptidase activities (AP) assay. Urine samples were collected for routine analysis by chemical tests. Aminopeptidase activities were determined by fluorometric methods in soluble (sol) and membrane-bound (mb) fractions of renal tissue, using arylamide derivatives as substrates. After the experimental period, the systolic blood pressure (SBP) values were similar in standard and VOO animals, and significantly lower than in the Bch group. At the same time, a significant increase in GluAP and IRAP activities were found in renal medulla of Bch animals. However, in VOO group the increase of GluAP activity in renal medulla was lower, while AspAP activity decreased in the renal cortex. Furthermore, the VOO diet also affected other aminopeptidase activities, such as TyrAP and pGluAP, related to the regulation of the sympathetic nervous system and the metabolic rate. These results support the beneficial effect of VOO in the regulation of SBP through changes in local AP activities of the kidney.

## 1. Introduction

High-fat diets (HFDs) are related to factors that condition the metabolic syndrome, such as the development of obesity and various pathologies such as hypertension, heart disease, stroke, atherosclerosis, diabetes, dyslipidemia, cancer and infertility [1,2,3,4,5]. Hypertension is the most prevalent factor in overweight people as a result of several mechanisms including hyperinsulinemia, insulin resistance, and increased sympathetic activity [6,7]. The ingestion of fatty acids from the diet induces changes in hemodynamic parameters and on hemorheology in the short and long term in animals and humans [6,7,8,9]. Overweight is a condition that affects body composition, damaging and modifying the aspects of the organs, with kidney injury being the most directly dependent on body weight [10].

Almost all the scientific evidence about a negative effect of fat on health refers to the source of saturated fatty acids and cholesterol. However, these results are not extensive to other types of lipid sources [11]. Previous studies have suggested that an increase in the degree of saturation of fatty acids in the diet causes an increase in total cholesterol concentrations in plasma, sympathetic activation [6,12] and systolic blood pressure values [11,13]. On the other hand, the reduction of saturated fatty acids in substitution by other sources of mono-/poly-unsaturated fatty acids attenuated the development of hypertension. These alterations depending on the degree of saturation of the fatty acid could be related to changes in the systemic or local renin angiotensin systems (RAS) [14,15] including the kidneys, due to their direct association with the development of hypertension [16,17,18].

Renin angiotensin system is one of the most important hormonal mechanisms involved in hemodynamic stability by regulating blood pressure, fluid balance, and electrolyte concentration. Until relatively recently, angiotensin II (Ang II) was considered the main peptide in the RAS. However, some of its metabolic derivatives, such as angiotensin III (Ang III), angiotensin IV (Ang IV) and angiotensin 2–10 (Ang 2–10), have also been shown to have important physiological functions [19,20]. The main active peptides of RAS (Ang II and Ang III) bind to angiotensin type 1 and 2 receptors (AT1 and AT2) with similar affinity [21,22]. In contrast, the ability of Ang IV to bind to AT1 and AT2 receptors is very low [21] but shows a high capacity to bind to the AT4 receptor, also described as insulin regulated aminopeptidase (IRAP) [23]. The binding of Ang IV to the AT4 receptor appears to play an important role in the regulation of local blood flow [24,25].

Different local RASs could be affected by changes in the degree of saturation of fat consumed in the diet. Within these local systems, angiotensin peptides are metabolized by various enzymes of the aminopeptidase (AP) family, also called angiotensinases, and previous results of our research group have shown that the activities of these peptidases are affected by the type of fatty acids consumed with the diet [5,14,25,26]. These APs are relevant in the control of blood pressure (BP) and renal function, participating in the regulation of the systemic and local RAS, but also like predictive renal injury biomarkers on the luminal surface of the renal tubule [27,28]. It has been suggested that several of the AP activities involved in angiotensin metabolism in renal and cardiovascular tissues of the rat may be modified depending on the amount and type of fat in the diet [15]. Among them, aspartyl-AP activities (AspAP; EC 3.4.11.21), responsible for the metabolism of Ang I to Ang 2–10; glutamyl-AP (GluAP or aminopeptidase A, APA; EC 3.4.11.7), metabolizes Ang II into Ang III; alanyl-AP (AlaAP or aminopeptidase M, APM; EC 3.4.11.2) and/or arginyl-AP (ArgAP or aminopeptidase B, APB; EC 3.4.11.6), responsible for the metabolism of Ang III to Ang IV and Ang 4–8; insulin-regulated AP (IRAP; 3.4.11.3), also called placental AP-leucyl (LAP), cystinyl-AP (CysAP), oxytokinase or vasopressinase, which as mentioned above, was identified as the binding site for the AT4 receptor Ang IV [29,30,31,32]. It has been suggested that the metabolizing activity of Ang II (GluAP) may be influenced by the composition of fatty acids in the diet and by the cholesterol content [5,14,15,33] directly or indirectly, and with an important role in the development of cardiovascular and kidney disorders.

Obesity-related kidney disease is linked to caloric excess promoting deleterious cellular responses. Accumulation of saturated free fatty acids in tubular cells produces lipotoxicity involving significant cellular dysfunction and injury [34]. Dietary fat affects the sympathetic nervous system (SNS) [12], where the sympathetic control of renal sodium tubular reabsorption is dependent on activation of the intrarenal renin–angiotensin system and the activation of the angiotensin II type 1 (AT1) receptor by angiotensin II. Increased angiotensin II levels in serum and urine have been observed in mice fed with an HFD. It is suggested that the intrarenal RAS activation may play an important role in diabetic kidney injury via mediating endoplasmic reticulum stress induced by saturated fatty acid [34].

Structural changes within the kidney, secondary to obesity, are important since fat deposition around the kidneys, together with increased abdominal pressure secondary to central obesity, has been suggested as an additional cause of disorder in the reabsorption of sodium [10]. Specifically, obesity causes renal vasodilation and glomerular hyperfiltration, which act as a compensatory mechanism to maintain sodium balance despite increased tubular reabsorption. Together with the increased blood pressure, metabolic abnormalities, as well as other factors such as inflammation, oxidative stress and lipotoxicity, can contribute to the exacerbation of kidney injury or dysfunction through a vicious cycle [35,36].

Catecholamines or adrenergic nerves innervating the kidney directly influence renal tubular function and, therefore, might participate in the regulation of sodium [37]. In the kidney, the catecholamine dopamine is mainly provided by renal tubules where it also encourages renin release and could inflict natriuresis, as well as inhibiting tubuloglomerular feedback [38,39]. It is known that levels for the neural marker, tyrosine hydroxylase, are significantly elevated in the hypertensive kidneys [40] and elevations in renal cortical catecholamines contribute to the development of hypertensive nephropathy [41].

In addition, these kidney disorders can cause internal and external effects that can be reflected in alterations of the levels of thyrotropin-releasing hormone, pGlu-His-Pro-NH2 (TRH). Similar to the liver [5], the kidney is a crucial organ for the breakdown of TRH. Kidney failure is a conditioner of physiological stress to inactivation of the hypothalamic hormone, compromising the regulation of thyroid hormones [42,43].

The Mediterranean diet (MD) is characterized by its main fat, virgin olive oil (VOO; from Olea europaea L.; Oleaceae), a natural oil with a high monounsaturated fatty acid (MUFA) content and a lot of bioactive components in its minor fractions. Virgin olive oil has demonstrated an important cardioprotective and antihypertensive role [44,45,46,47,48]. To analyze the beneficial effects of VOO, as an important component of the MD, compared to a saturated fat, typical of Western diets, in this work we determine the evolution of systolic blood pressure (SBP) and several angiotensinases activities in the kidney of male Wistar rats fed during six months with a diet supplemented with virgin olive oil (20%) or with a diet supplemented with butter (20%) plus cholesterol (0.1%) compared to a standard diet (S). In addition, several physiological parameters, such as water intake or urine volume were quantified.

## 2. Results

### 2.1. Systolic Blood Pressure, Water Intake and Diuresis

The results obtained from SBP values indicated that during the first two months no significant differences were observed between the three groups of animals. However, from the third month to the end of the experimental period, a marked increase was observed for the animals that were consuming the Bch diet. Interestingly, there were no differences between the animals fed with S and VOO diets, even when the VOO diet contained a percentage of fat similar to Bch diet. No significant differences were observed for the data of water intake, diuresis and the relation between both values, despite a slight but nonsignificant decrease in the urine excretion (mL/100 g body weight) in the group of rats fed VOO diet (Figure 1). Although not significant, there was also a decrease in water intake for VOO and Bch groups (Figure 1).

### 2.2. Angiotensinase Activities

The striated texture of the medullary parenchyma is due to the predominance of collecting tubules and uriniferous ducts in the pyramids (Figure 2). The granular texture of the cortical parenchyma is due to the fact that the glomeruli of the nephrons are located there (Figure 2). This anatomical distinction of the renal parenchyma also determined the differences in the enzymatic activities, independently of the experimental diets on their soluble or membrane-bound fractions (Figure 2). In the renal cortex (Figure 2) it was observed that the VOO diet reduced significantly, even below the S diet without being significant, the membrane activity of AspAP. In the renal medulla (Figure 2), significant differences were observed for the soluble activities of GluAP and IRAP. GluAP activity was higher with the two HFDs compared to S diet. However, it was also significantly lower with the VOO diet compared to the Bch diet. The increase in soluble GluAP activity was also parallel with the increase in IRAP activity, but in this case, a significant difference compared to standard diet was achieved only with Bch diet.

### 2.3. Tyrosine Aminopeptidase Activity

The mean values of TyrAP activity in the soluble fraction of the renal medulla were lower with the VOO diet compared to the Bch diet (Figure 3A). In the membrane fraction of the renal medulla, the VOO diet also produced a significant reduction in TyrAP compared to the S diet (Figure 3A).

### 2.4. Pyroglutamyl Aminopeptidase Activity

The degrading activity of TRH was decreased significantly by the HFDs in the membrane fraction of the renal cortex. However, the VOO diet significantly reduced this activity in the soluble fraction of the renal medulla, achieved statistical significance only compared to Bch diet (Figure 3B).

### 2.5. Determination of Renal Pathological Changes Using Urine Test Strip

The test strip for urinalysis are in vitro diagnostic dipsticks to determine different parameters allocated in urine, among them: glucose, nitrite, protein, bilirubin, ketone, pH, density, and leukocytes. The results obtained on these test strips were read visually and can supply both semiquantitative and qualitative determination (Table 1). For nephropathy screening associated to disorders such diabetic nephropathy, hypertension, cardiovascular diseases, and chronic kidney disease, the measurement of urinary albumin is considered as a cost-effective procedure which could minimize disease progression [49].

Urinalysis showed no glucose signal in the samples (data not shown). The determination of protein in urine detected traces within the normal range of albuminuria in both HFDs than in S diet. However, there was a significant interaction in nitrite levels by the dominant diet effect, but there were no differences in nitrite levels between the experimental diets. Compared to control animals, VOO and Bch groups showed a lower ketone and pH value in urine, but within the reference range. The densities (specific gravity) were significantly higher in both HFDs. Bilirubin and leukocytes were significantly lower for VOO diet compared to S diet.

## 3. Discussion

A local renin angiotensin system (RAS) has been widely characterized in the kidney. This system acts locally as a paracrine mechanism, regulating both the glomerular flow rate and the excretion of water and electrolytes. Moreover, the components of local RAS are able to reach the peripheral circulation and act as an endocrine system [50]. Renal diseases have been related to obesity and hypertension [51] and to alterations in kidney physiology, characterized by proteinuria, inflammation, endothelial dysfunction and tubular atrophy. Tubular damage results in a low-grade proteinuria, typically of low molecular weight proteins. Glomerular damage results in the loss of selectivity during protein filtration exacerbate by the hyperfiltration [52]. High-fat diet, by itself, induces alterations in kidney functions, with an increase in serum creatinine and a decrease in urine volume [53]. Hypertension combined with obesity also alters kidney physiology. Urinary albumin excretion is increased in obese spontaneously hypertensive rats (SHR) compared to control animals, and HFDs reduce creatinine clearance and affect the renal function in SHR animals [54]. Our result did not demonstrate changes in the urine output or albuminuria in VOO or Bch diets compared to standard diet, indicating no renal damage. The development of hypertension and renal damage associated to HFDs also have a link with changes in gut microbiota [55,56,57]. However, the outcomes of these cardiometabolic and renal diseases could be improved by the use of pro- and prebiotics with the diet [58,59].

Tubular epithelial cells synthesize metabolites able to attract inflammatory cells into the renal interstitium [52,60]. Nitric oxide (NO) is metabolized to nitrate and nitrites, which are excreted in the urine. Previous studies have shown that HFDs diminish the urinary excretion of nitrite, with or without Ang II infusion [60]. These results suggest that short/long-term HFD increases oxidative stress and afterward a decrease of NO bioavailability and urinary nitrate/nitrite excretion [54,60,61]. However, our data indicate a nonsignificant higher nitrite excretion in urine with the HFDs than with standard diet. Cao et al. (2012) reported lower levels of eNOS and higher levels of iNOS in the renal cortex of SHR fed a HFDs, although did not find differences in urine nitrites.

Urine levels of bilirubin were lower in the VOO group, indicative of a decrease in liver heme oxygenase activity (HO). SHR fed an HFD also decreased the bilirubin in urine [54]. The heme-HO system is a key mechanism in the defense against oxidative stress, beside decrease SBP in hypertensive rats and increase adiponectin with suppression of inflammatory cytokines in obese and non-obese animal models [54].

Hypertensive models fed an HFD increase macrophage infiltration in kidney [54]. Our animals fed the VOO diet present lower numbers of leukocytes in urine, indicating a decrease in the number of macrophages infiltrated in the renal parenchymal.

As indicated previously, the source of fat in the diet has demonstrated to be decisive in the development of metabolic syndrome, obesity and hypertension [62,63,64]. Changes in systemic and local RAS play an important role in this process [4,5,15,65,66]. The increase in the blood pressure values results in an increase of the renal artery pressure, triggering a high rate of filtration and increasing the delivery of sodium and water to the distal nephron, where the macula densa is located [67]. The pressure natriuresis mechanism reduces the reabsorption of sodium and water, to recover the homeostasis of extracellular fluid volume and normal values of systolic blood pressure [68]. Interestingly, only the animals fed the Bch diet showed a significant increase in the systolic blood pressure after two months of the experimental procedure. However, animals of two HFDs diminished their water intake at the same time that their urine excretion did not increase. Thus, the rise in SBP in the Bch diet was not due to an increase of the effective circulating volume (ECV), and other mechanisms seem to be implicated. It is known that the HFDs induce a stimulatory effect on the norepinephrine turnover, consequent with an increase of sympathetic activity, the blood pressure values and the production of Ang II in the kidneys [6,12,34]. Several studies from our research group have demonstrated previously the important role of local kidney RAS in different models of hypertension [16,25,69,70,71,72,73]. The results of our study confirm this idea, with significant changes in several renal angiotensinase activities in both VOO and Bch diets, thought the differences with standard diet were more remarkable in Bch than in the VOO diet, and these differences were located mainly in renal medulla. The administration of a diet enriched with virgin olive oil is able to delay the outset of hypertension in SHR rats, and this effect is related to changes in several aminopeptidase activities [15]. Our results also demonstrated the beneficial effect of virgin olive oil on the control of blood pressure. However, the changes found in renal aminopeptidase activities, a decrease of AspAP activity in renal cortex in this group, are not in agreement with the previous results found in SHR animals fed with a diet enriched with virgin olive oil. However, these differences could be explained by the use of different animal models.

The decrease of AspAP activity in the renal cortex of animals fed with VOO diet could be related to lower levels of Ang 2–10, a vasodilator peptide. However, the GluAP activity is higher in the renal medulla of these animals compared to controls, indicating upper metabolism of Ang II to Ang III, a vasodilator peptide. Taken together, this result indicates a higher flow in the renal medulla, which can have an effect on sodium and water excretion, with a long-term influence on arterial blood pressure regulation. Conversely, in Bch animals the changes in aminopeptidase activity were located only at renal medulla level. The GluAP activity was higher than in control animals, even higher than in the VOO group, indicating a rapid metabolism of Ang II to Ang III. The high metabolism of Ang II to Ang III in renal medulla is consistent with previous studies that showed the importance of this peptide in the physiology of renal medulla [74]. The intrarenal Ang III, not Ang II or Ang (1–7), induces natriuresis via activation of AT2 receptor [75], and this peptide could be a homeostatic mechanism to control the increase in SBP. In fact, the increase of Ang III in kidney has been proposed as a treatment for hypertension [75]. Our results support the hypothesis that the increase in SBP correlated with an activation of renal RAS [15,76,77], and this activation occurs mainly in renal medulla [15].

The increase in the activity of GluAP in renal medulla of Bch animals was parallel with an increase in IRAP activity. The insulin regulated aminopeptidase (IRAP) is also the AT4 receptor, which bonds to Ang IV, a peptide release by the metabolism of Ang III. IRAP colocalizes with GLUT4 in specialized vesicles, where it plays a tethering role. In the cells of the inner medulla, IRAP plays a localized role in the regulation of vasopressin bioactivity, and the AQP-2 levels are two-fold higher in IRAP knockout mice [78].

It is well known that during hypertension, Ang II send signals to sympathetic excitatory centers [41,79]. On the other hand, it has been suggested that both high fat and high carbohydrates diets stimulate peripheral α1 and β adrenergic receptors thereby leading to the elevation of sympathetic activity and hypertension [79,80]. In kidney, a bilateral sympathetic denervation is able to prevent hypertension and sodium retention associated with obesity [39]. Besides that, upregulated hypothalamic and renal tyrosine hydroxylase have been identified in obese and/or hypertensive rats [39] and the development of hypertension in cafeteria-fed rats is associated with changes in the renal subtypes of adrenergic α2 receptors [12]. In the kidneys, the noradrenergic catecholamine is regulated by the activity of the renal sympathetic nerve. However, the renal dopamine is mainly generated by the uptake of L-DOPA by the renal tubules [12]. Animals fed the VOO diet showed significantly lower levels of TyrAP activity compared with control and Bch diets. The lower release of Tyr into the kidney could mean a decrease in the substrate for the synthesis of catecholamines. Dopaminergic actions in the kidney are not limited to sustaining sodium homeostasis. Dopamine may increase GFR, by post glomerular efferent arteriolar constriction, and modulate renin expression as well as Ang II, also being able to control sodium excretion and SBP [12].

The dopaminergic system is important in the control of renal function and homeostasis, adapting kidneys to the different physiological or pathological situations, and maintaining the sodium homeostasis, the extracellular volume, and blood pressure values. On the other hand, several strategies for the treatment of hypertension blocked this system that led to renal vasoconstriction [81]. A close relationship has been established between Ang II and the dopaminergic system, being that Ang II is able to diminish the excretion of sodium in the presence of exogenous dopamine [82]. Therefore, in VOO animal’s diet it seems to be able to increase the metabolism of Ang II to Ang III, through the high activity of GluAP, and decrease the synthesis of dopamine due to the lower levels of TyrAP.

The enzyme pGluAP is involved in the inactivation of GnRH and TRH [83]. Thyrotropin-releasing hormone (TRH) participates in the regulation of blood pressure in diverse animal models, independently of the thyroid status. Transgenic mice that overexpress TRH show an increase in blood pressure accompanied by changes in body weight and food consumption mediated by a higher sympathetic overflow, indicating the participation of TRH in cardiovascular and body weight regulation [84]. A low activity of soluble pGluAP in renal cortex and medulla has been suggested in dehydrated rats [85]. However, previous results showed no differences in kidney pGluAP activity of different animal models of renovascular hypertension [69]. The efficiency of treatments applied as hydro saline challenges to study alterations in body fluid volume and osmolality (effects on hematocrit, plasma protein concentration, plasma osmolality and body weight), did not show differences in pGluAP activity in the kidney [86]. Previous results from our laboratory have demonstrated the influence of different fat sources on pGluAP activity in several tissues of mice [33] and rats [5]. The results of the present study confirm the role of dietary fat on pGluAP activity, with lower levels in the membrane fraction of renal cortex of animals fed both HFDs, and high activity in the soluble fraction of renal medulla, but only in the VOO group.

At present, the possible mechanism that explains the protective role of virgin olive oil on hypertension and renal damage it is not totally explained. This effect could be related to the special fatty acid composition of virgin olive oil (rich in monounsaturated fatty acid) and changes in the cellular membranes composition that could lead to modifications in fluidity and enzyme activities. On the other hand, it is also possible that the protective role could be related to the antioxidant and anti-inflammatory components of the virgin olive oil [87]. Indeed, a decrease in the urine 8-isoprotanes has been described in SHR rats fed with a diet enriched with virgin olive oil [15]. The minor components of virgin olive oil also have demonstrated other interesting bioactive properties, such as the antihypertensive effects of their low molecular weight peptides [88].

These results are clinically relevant, because they demonstrated that the inclusion of virgin olive oil in the diet is able to prevent the development of arterial hypertension compared to a source of saturated fatty acids and cholesterol. This effect implicates mechanisms that modify the metabolism of renal angiotensin peptides, and the decrease of the catecholamine-forming precursors.

## 4. Materials and Methods

### 4.1. Animals and Diets

Adult male Wistar rats were purchased from Harlan Interfauna Ibérica S.A. (Barcelona, Spain). The experiments were approved according to the Institutional Animal Care and Use Committee of the University of Jaén, with code project number P.I.UJA_2005_acción 14 (1 January 2006). Rats had free access to experimental diets and water during 24 weeks, and they were maintained under a controlled temperature (20–25 °C) and humidity (50 ± 5%) environment, in a 12 h light/dark cycle. At the beginning of the study, the mean body weight and age of animals were ±495 g and six months old, respectively. Experimental procedures for animal use and care were in accordance with European Communities Council Directive 2010/63/UE and Spanish regulation RD 53/2013, and the study was approved by the Institutional Animal Care and Use Committee of the University of Jaén. Rats were randomly assigned into three groups: in the standard diet (S, *n* = 6) group, rats were fed with a commercial chow for laboratory rodents (Panlab, Barcelona, Spain) whose nutritional composition was 16.5% protein, 3% total fat, 60% carbohydrates (nitrogen-free extract (NFE)), 5% minerals, and 4% fiber. The other two diets were HFDs. One group of rats was fed with the S diet supplemented with 20% of virgin olive oil (VOO, *n* = 5), composed of a total fat content (%) of ω-9-monounsaturated fatty acid (oleic acid, C18:1) 75.5%, saturated fatty acid (palmitic acid, C16:0) 11.5%, and ω-6-polyunsaturated fatty acid (linoleic acid, C18:2) 7.5%). The second group of rats was fed with the S diet supplemented with 20% butter plus cholesterol (0.1%) (Bch, *n* = 5), composed of a total fat content (%) of monounsaturated fatty acid (C18:1) 29%, saturated fatty acid (C16:0 y C18:0) 62%, polyunsaturated fatty acid (C16:0) 4%, and short and medium chain fatty acids (C4–C14). The Bch diet was supplemented with cholesterol (0.1%), in order to reach the average cholesterol content of the Western diet. The HFD diets were isocaloric, 1848 KJ/100 g and 1827 KJ/100 g for EVOO and Bch diets, respectively, compared to 1392 KJ/100 g S diet. The water and food intake of each group were measured during the balance periods.

After obtaining the blood samples, at the end of the experimental period the animals were perfused with a saline solution (9‰) through the left cardiac ventricle under Equithensin anesthesia (2 mL/kg body weight). The left kidney was dissected, and the samples were immediately placed in liquid nitrogen to separate into two parts (cortex and renal medulla), as previously described [89], and were frozen at −80 °C until their use.

### 4.2. Systolic Blood Pressure, Water Intake and Diuresis Quantification

The systolic blood pressure (SBP) measurement was performed with the Letica-5000 measurer in unanesthetized animals, using the “tail-cuff” pneumatic plethysmography method as previously described [89,90]. The animals were placed in plastic holder and warmed to 37 °C for 15–20 min. Air was blown into the occlusion cuff until 250 mmHg was reached, the reference value for occlusion of the caudal artery. The occluder cuff was then abruptly deflated until an indicator light on the meter was signaled, corresponding to the SBP value and indicating the appearance of increased and turbulent blood flow as a result of hyperemia. At least seven measurements of SBP were carried out in each session, considering the mean of the three lowest values within a range of 5 mmHg as the SBP value. All measurements were performed during the same period of the day (10 a.m–12 a.m.), knowing SBP has diurnal variations.

Individual metabolism cages were used at the second and sixth experimental month, in order to obtain values of food and water intake, as well as urine excretion for each group. Urine samples were analyzed using a colorimetric reagent strip (Multistix, Siemens, Tarrytown, NY, USA) for the detection of leukocytes, nitrites, pH, protein, density, glucose, ketones, bilirubin.

### 4.3. Aminopeptidases Activities Assay

In order to obtain the soluble fraction, the tissue samples were homogenized with 0.5 mL of 10 mM Tris-HCl buffer (pH 7.4) and ultracentrifuged at 100,000× *g* for 30 min at 4 °C. The resulting supernatants were used to measure the enzymatic activities corresponding to the soluble fraction, and the protein content. To solubilize the membrane proteins, the pellets were rehomogenized in 10 mM Tris-HCl buffer (pH 7.4) with 1% Triton-X-100, and ultracentrifuged at 100,000× *g* for 30 min at 4 °C. The supernatants were kept for at least 4 h at 4 °C and shaken with SM-2 biobeads, in order to remove the detergent used to solubilize the membrane proteins. The resulting samples were used to measure the enzymatic activities of the membrane-bound fraction, and the protein content.

The enzymatic activities of soluble and membrane-bound fractions of AlaAP, ArgAP, AspAP, IRAP, GluAP, pGluAP and TyrAP, were measured by a fluorimetry assay using as substrates aminoacyl-β-Naphthylamides (aa-β-NA): L-Ala-β-NA, L-Arg-β-NA, L-Asp-β-NA, L-Cys-β-NA, L-Glu-β-NA, L-pGlu-β-NA, and L-Tyr-β-NA, respectively, according to the methods of different authors [91,92,93] modified by Prieto and Ramírez [71,94]. The 96-well black plates were used, and 10 µL of the supernatant sample was pipetted into each well and incubated for 30 min at 37 °C in 100 µL of substrate solutions. After 30 min of incubation, the enzymatic reactions were stopped by adding 100 µL of 0.1 M acetate buffer (pH 4.2). The β-NA released as a result of the enzymatic activity, was quantified fluorometrically at 412 nm emission with an excitation of 345 nm. Each determination was made in triplicate. The soluble and membrane-bound activities were expressed as pmoles of L-Ala-β-NA, L-Arg-β-NA, L-Asp-β-NA, L-Cys-β-NA, L-Glu-β-NA, L-pGlu-β-NA and L-Tyr-β-NA hydrolyzed per minute and per mg of protein (pmol aa-β-NA/min/mg prot).

### 4.4. Protein Measurement

For the quantification of proteins, the method of Bradford [95], based on the emission of light at 595 nm, was used. Bovine serum albumin (BSA) was used as a standard.

### 4.5. Statistical Analysis

Statistical analysis was performed using one-way ANOVA, followed by Tukey’s post-hoc test for multiple comparisons. When the normality test failed, Kruskal–Wallis One Way Analysis of Variance on Ranks was performed. Significant differences were estimated with Sigmaplot v11.0 software (Systat Software, Inc., San José, CA, USA), and *p*-values below 0.05 (*p* < 0. 05) were considered statistically significant. All data are presented as mean ± standard error of the mean (SEM). Quantitative variables that were meant to be nonparametric univariate statistics, median and interquartile range (IQR) were used.

## 5. Conclusions

In conclusion, this study presents new data regarding the aminopeptidases’ regulation on blood pressure control and local renal RAS. The intake of saturated fat raised systolic blood pressure related to an increase of GluAP, and IRAP activity in renal medulla. However, extra virgin olive oil presents a protective effect on systolic blood pressure, showing a slight increase in GluAP activity in the renal medulla, lower than in Bch diet. Virgin olive oil seems present an indirect effect on the sympathetic system and the metabolic activity in kidney through changes on the TyrAP and pGluAP activities, reducing the release of tyrosine residues necessary for the formation of new catecholamines and the degradation of TRH.

## Figures and Tables

**Figure 1 ijms-22-05388-f001:**
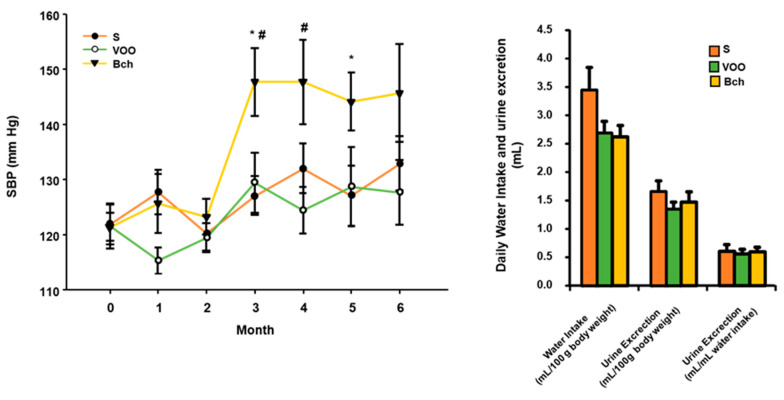
Means values ± standard errors of systolic blood pressure (SBP) expressed as mmHg, water intake and urine excretion, expressed as mL/100 g BW and the relationship between urine excretion and water intake. S: standard diet, VOO: virgin olive oil diet, Bch: butter plus cholesterol diet. * indicates significant differences between VOO or Bch vs. S, * *p* < 0.05. # indicates significant differences between VOO and Bch, # *p* < 0.05.

**Figure 2 ijms-22-05388-f002:**
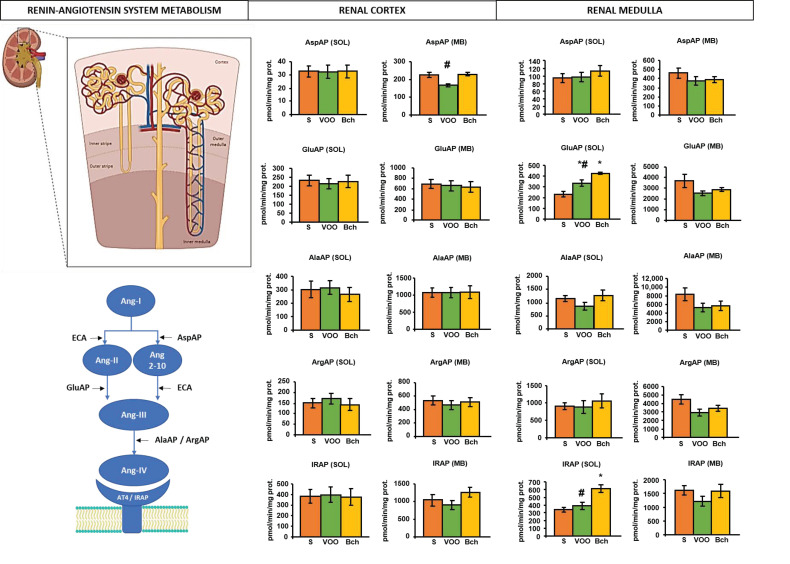
Anatomical representation of the renal parenchyma and partial scheme of the classic axis of the renin–angiotensin system, displaying the metabolic steps in which angiotensinase activities are involved. Mean values ± standard errors of aspartyl-, glutamyl-, alanyl-, arginyl- and insulin-regulated aminopeptidase activity (AspAP, GluAP, AlaAP, ArgAP, IRAP) in soluble (SOL) and membrane-bound (MB) fractions of renal cortex and medulla. Values are expressed as pmol/min/mg prot. * indicates significant differences between VOO or Bch vs. S, * *p* < 0.05. # indicates significant differences between VOO and Bch, # *p* < 0.05.

**Figure 3 ijms-22-05388-f003:**
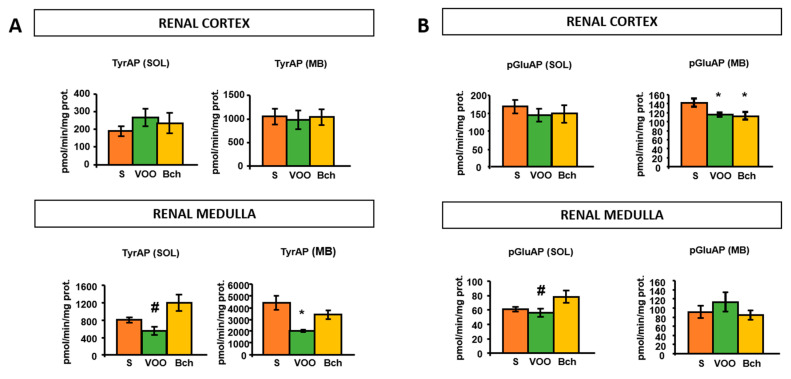
Mean values ± standard errors of (**A**) tyrosyl aminopeptidase activity (TyrAP) and (**B**) pyroglutamyl aminopeptidase activity (pGluAP) in soluble (SOL) and membrane-bound (MB) fractions of renal cortex and medulla, expressed as pmol/min/mg prot. * indicates significant differences between VOO or Bch vs. S, * *p* < 0.05. # indicates significant differences between VOO and Bch, # *p* < 0.05.

**Table 1 ijms-22-05388-t001:** Test range of urinary measurement patterns.

Median; (Interquartile Range)	S Diet	VOO Diet	Bch Diet	*p* Value
Nitrite (mg/dL)	0; (0–0)	0; (0–0)	0; (0–1)	0.038
Protein (mg/dL)	2; (1.5–2)	1.5; (1–2)	1.5; (1–2)	0.438
Bilirrubin (mg/dL)	1; (0.5–1)	0; (0–0)	1; (0–1)	0.004 S vs. VOO
Ketone (mmol/L)	1; (0.5–1)	0; (0–0.5)	0; (0–0.5)	≤0.001 S vs. VOO S vs. Bch
pH	8; (7.75–8.5)	6.5; (6.5–7.75)	6.25; (6–7.25)	0.001 S vs. VOO S vs. Bch
Density (SG)	1002.5; (1000–1010)	1027.5; (1010–1030)	1030; (1017.5–1030)	0.001 S vs. VOO S vs. Bch
Leukocytes (WBC/µL)	0.5; (0.5–1)	0; (0–0)	0.5; (0–0.5)	0.008 S vs. VOO

Note: Values represent median and interquartile range (IQR) of urinary measurement parameters (nitrite, protein, bilirubin, ketone, pH, density, leukocytes). *p* < 0.05, significance was observed; *n* = 5–6 for each group. S: standard chow diet; VOO: 20% virgin olive oil diet; Bch: 20% butter plus 1% cholesterol diet; SG: specific gravity; WBC: white blood cells.

## Data Availability

Not applicable.

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
