# Peer review of "Effects of Virgin Olive Oil on Blood Pressure and Renal Aminopeptidase Activities in Male Wistar Rats"

_ijms, 2021, doi:10.3390/ijms22105388_

Round 1

Reviewer 1 Report

This is a well-written original paper on an emerging topic of the effects of virgin olive oil on blood pressure and renal Ami-2 nopeptidase activities. Authors provided strong evidence of changes caused by virgine olive oil diet and standard diet enriched with butter on to systolic blood pressure, angiotensinase activities, tyrosine aminopeptidase activity, pyroglutamyl aminopeptidase activity.

In general, manuscript written in a proper style and with minor correction deserves publication.

Minor spell check recommended

(i.e. L19 "was" -> "were", L328 The rats ; L335 decapitalize "In", etc.; check for minor typos)

Lack of modern literature can be smoothed with including of the following publications on the topic of probiotic supplementation in your references list:
https://doi.org/10.3390/microorganisms8081225 (2020)
https://doi.org/10.14814/phy2.14610 (2020)
https://doi.org/10.1186/s12866-021-02099-0 (2021)

Best wishes!

Author Response

Authors’ Response to the Reviewer’s comments

Journal:               International Journal of Molecular Sciences

Title of Paper:     Effects of Virgin Olive Oil on Blood Pressure and Renal Aminopeptidase Activities in Male Wistar Rats

Authors:              Germán Domínguez-Vías, Ana Belén Segarra, Manuel Ramírez-Sánchez, Isabel Prieto

Date Sent:           1 May 2021

Dear Reviewer,

We appreciate your time and efforts in reviewing our article. All the issues indicated in the comments from Reviewers have been addressed.

We have corrected the typographical errors in the manuscripts and have amended the lack abbreviations. The article was checked using the writing assistant tool. Major changes, made according to the Reviewers’ suggestions, are marked in red in the text.

Yours faithfully,

Germán Domínguez-Vías

Reviewer 1

  • This is a well-written original paper on an emerging topic of the effects of virgin olive oil on blood pressure and renal Ami-2 nopeptidase activities. Authors provided strong evidence of changes caused by virgine olive oil diet and standard diet enriched with butter on to systolic blood pressure, angiotensinase activities, tyrosine aminopeptidase activity, pyroglutamyl aminopeptidase activity.

Thank you. Indeed, that is a good summary that explains the content of the manuscript.

  • In general, manuscript written in a proper style and with minor correction deserves publication.

We agree with your assessment. We have corrected minor typographical errors.

  • Minor spell check recommended
    (i.e. L19 "was" -> "were", L328 The rats ; L335 decapitalize "In", etc.; check for minor typos)

As you have pointed out, the typological mistake has been corrected.

  • Lack of modern literature can be smoothed with including of the following publications on the topic of probiotic supplementation in your references list:
    https://doi.org/10.3390/microorganisms8081225 (2020)
    https://doi.org/10.14814/phy2.14610 (2020)
    https://doi.org/10.1186/s12866-021-02099-0 (2021)

We reiterate our thanks to the reviewer for improve this manuscript with new bibliography that links renal alterations with changes in the microbiota. A paragraph addressing this idea has been added between lines 217 to 219:

“The development of hypertension and renal damage associated to HFD also have link with changes in gut microbiota [Prieto et al., 2018; Jiang et al., 2020; Zhu et al., 2021]. However, the outcomes of these cardiometabolic and renal diseases could be improved by the use of pro- and prebiotic with the diet [Karaduta et al., 2020; Lkhagva et al., 2021]”.

Prieto I, Hidalgo M, Segarra AB, Martínez-Rodríguez AM, Cobo A, Ramírez M, Abriouel H, Gálvez A, Martínez-Cañamero M. Influence of a diet enriched with virgin olive oil or butter on mouse gut microbiota and its correlation to physiological and biochemical parameters related to metabolic syndrome. PLoS One. 2018 Jan 2;13(1):e0190368. doi: 10.1371/journal.pone.0190368. PMID: 29293629; PMCID: PMC5749780.

Zhu Y, Liu Y, Wu C, Li H, Du H, Yu H, Huang C, Chen Y, Wang W, Zhu Q, Wang L. Enterococcus faecalis contributes to hypertension and renal injury in Sprague-Dawley rats by disturbing lipid metabolism. J Hypertens. 2021 Jun 1;39(6):1112-1124. doi: 10.1097/HJH.0000000000002767. PMID: 33967216.

Reviewer 2 Report

1) Table 1 should have footnotes and also abbreviations. 

2) Please provide the IRB number of study approval for this study.

3) The authors should discuss what the clinical utility of these findings could be 

4) There should be more discussion of a possible mechanism 
5) Explain what sort of future study might take us closer to a clinical utility

Author Response

Authors’ Response to the Reviewer’s comments

Journal:               International Journal of Molecular Sciences

Title of Paper:     Effects of Virgin Olive Oil on Blood Pressure and Renal Aminopeptidase Activities in Male Wistar Rats

Authors:              Germán Domínguez-Vías, Ana Belén Segarra, Manuel Ramírez-Sánchez, Isabel Prieto

Date Sent:           1 May 2021

Dear Reviewer,

We appreciate your time and efforts in reviewing our article and all the valuable contributions in order to improve the quality of our work, as well as the discussion of several aspects of clinical relevance. Major changes, made according to the Reviewers’ suggestions, are marked in red in the new version of the manunscript.

Yours faithfully,

Germán Domínguez-Vías

Reviewer 2

  • Table 1 should have footnotes and also abbreviations. 

We appreciate the reviewer's recommendation. We have amended Table 1, adding footnotes and abbreviations (lines 198 to 200).

  • Please provide the IRB number of study approval for this study.

The request has been included in Materials and Methods (lines from 347 to 349): “The experiments were approved according to the Institutional Animal Care and Use Committee of the University of Jaén, with code project number P.I.UJA_2005_acción 14”.

Likewise, it appears in the "Institutional Review Board Statement" section (lines from 445 to 446):

“and the study was approved by the Institutional Animal Care and Use Committee of the University of Jaén (project number P.I.UJA_2005_acción 14).

  • The authors should discuss what the clinical utility of these findings could be.

Authors agree with the reviewer about the relevance of clinical implications in this study.

We have added a paragraph that shows the importance of our results in this sense between lines 339 to 343:

“These results are clinically relevant, because, demonstrated that the inclusion of virgin olive oil in the diet is able to prevent the development of arterial hypertension compared to a source of saturated fatty acids and cholesterol. This effect implicates mechanisms that modify the metabolism of renal angiotensin peptides, and the decrease of the catecholamine-forming precursors”.

  • There should be more discussion of a possible mechanism 

We add a possible mechanism between the lines 329-338:

“At present, the possible mechanism that explain the protective role of virgin olive oil on hypertension and renal damage it is not totally explained. This effect could be related to the special fatty acid composition of virgin olive oil (rich in monounsaturated fatty acid) and changes in the cellular membranes composition that could lead to modifies in fluidity and enzyme activities. On the other hand, also is possible that the protective role could be related to the antioxidant and anti-inflammatory components of the virgin olive oil [https://doi.org/10.3390/foods10040839]. Indeed, a decrease in the urine 8-isoprotanes has been described in SHR rats fed with a diet enriched with virgin olive oil [15]. The minor components of virgin olive oil also have demonstrated others interesting bioactive properties, such as the antihypertensive effects of their low molecular weight peptides [http://dx.doi.org/10.3390/nu12010271]”.

5) Explain what sort of future study might take us closer to a clinical utility

Further studies are needed in order to stablish if the effects of virgin olive oil are associated to saponifiable or unsaponifiable fractions. For example, analyse the effects on different animals models of diets supplemented with refine olive oil (without the minor components), or with an olive leaf extract rich in phenolic compounds.

Both studies would help to more precisely define the search for an active biocomponent with an antihypertensive capacity. The discovery of therapeutic targets as promising tools to correct hypertension continues to be studied.

Round 2

Reviewer 2 Report

The authors have responded appropriately. This is an interesting paper with increased scientific soundness after their corrections.